# A Comparative Study of the Hepatoprotective Effect of *Centella asiatica* Extract (CA-HE50) on Lipopolysaccharide/d-galactosamine-Induced Acute Liver Injury in C57BL/6 Mice

**DOI:** 10.3390/nu13114090

**Published:** 2021-11-15

**Authors:** Woojae Hong, Jeon Hwang-Bo, Hyelin Jeon, Minsung Ko, Joongyeon Choi, Yong-Joon Jeong, Jae-Hyun Park, Inhye Kim, Tae-Woo Kim, Hyunggun Kim, Se-Chan Kang

**Affiliations:** 1Department of Biomechatronic Engineering, Sungkyunkwan University, Suwon 16419, Gyeonggi, Korea; woojaehong94@gmail.com (W.H.); minsungko413@gmail.com (M.K.); joongyeonchoi96@gmail.com (J.C.); 2Department of Oriental Medicine Biotechnology, College of Life Sciences, Kyung Hee University, Yongin 17104, Gyeonggi, Korea; hbj3286@khu.ac.kr; 3Research Institute, GENENCELL Co., Ltd., Yongin 16950, Gyeonggi, Korea; jeonhl0219@genencell.co.kr (H.J.); jeyoon@genencell.co.kr (Y.-J.J.); chadol9270@genencell.co.kr (J.-H.P.); ihyekim@naver.com (I.K.); twkim@genencell.co.kr (T.-W.K.)

**Keywords:** *Centella asiatica*, acute liver failure, hepatoprotective effect, antioxidant

## Abstract

Acute liver failure (ALF) refers to the sudden loss of liver function and is accompanied by several complications. In a previous study, we revealed the protective effect of *Centella asiatica* 50% ethanol extract (CA-HE50) on acetaminophen-induced liver injury. In the present study, we investigate the hepatoprotective effect of CA-HE50 in a lipopolysaccharide/galactosamine (LPS-D-Gal)-induced ALF animal model and compare it to existing therapeutic silymarin, *Lentinus edodes* mycelia (LEM) extracts, ursodeoxycholic acid (UDCA) and dimethyl diphenyl bicarboxylate (DDB). Serum aspartate aminotransferase (AST) and alanine aminotransferase (ALT) levels were decreased in the CA-HE50, silymarin, LEM, UDCA and DDB groups compared to the vehicle control group. In particular, AST and ALT levels of the 200 mg/kg CA-HE50 group were significantly decreased compared to positive control groups. Lactate dehydrogenase (LDH) levels were significantly decreased in the CA-HE50, silymarin, LEM, UDCA and DDB groups compared to the vehicle control group and LDH levels of the 200 mg/kg CA-HE50 group were similar to those of the positive control groups. Superoxide dismutase (SOD) activity was significantly increased in the 100 mg/kg CA-HE50, LEM and UDCA groups compared to the vehicle control group and, in particular, the 100 mg/kg CA-HE50 group increased significantly compared to positive control groups. In addition, the histopathological lesion score was significantly decreased in the CA-HE50 and positive control groups compared with the vehicle control group and the histopathological lesion score of the 200 mg/kg CA-HE50 group was similar to that of the positive control groups. These results show that CA-HE50 has antioxidant and hepatoprotective effects at a level similar to that of silymarin, LEM, UDCA and DDB, which are known to have hepatoprotective effects; further, CA-HE50 has potential as a prophylactic and therapeutic agent in ALF.

## 1. Introduction

Acute liver failure (ALF) is an abrupt hepatocyte injury causing massive liver function loss in patients with no prior liver disease [1]. One of the leading causes of death in patients with ALF is systemic complications, which result in cardiovascular collapse, kidney failure and cerebral edema. The underlying etiologies of ALF include viral infections, drug-induced liver injury and hepatic ischemia [2]. To evaluate the degree of liver injury, serum biochemical indicators (AST, aspartate transaminase; ALT, alanine transaminase; LDH, lactate dehydrogenase; TG, triglyceride; TC, total cholesterol) and liver tissue oxidation indicators (CAT, catalase; SOD, superoxide dismutase) are used [3]. AST, ALT and LDH enzyme activities are low under normal conditions, but, when the liver tissue is damaged, AST, ALT and LDH are released into the bloodstream, increasing serum concentrations and increasing their activities. Thus, measurements of serum levels of AST and ALT are effective for the diagnosis of liver damage [4]. Liver injury can lead to the transfer of fatty acids to the liver, increasing the TG content in the liver, whereas TC reacts to lipid peroxidation in the liver and TC levels in the body increase [5]. The oxidative stress induces an immune response or an inflammatory reaction in several types of cells in the liver, causing liver damage and the balance of the antioxidant defense system is broken, leading to a variety of diseases [6,7]. To remove ROS, antioxidant enzymes such as CAT, glutathione peroxidase (GPx), SOD and glutathione reductase (GR), and non-enzymatic antioxidants such as glutathione (GSH) work in the liver cells [8]. Many studies have been conducted to test treatments for ALF, but liver transplantation remains the best treatment strategy. However, limited tissue donations, high liver transplantation requirements and high incidence of post-transplant complications necessitate research into alternative therapies such as ALF prophylactic and therapeutic drugs.

Therefore, research on alternative treatments is being actively conducted and, in particular, treatment studies using natural products with lower toxicity or side effects and higher stability than synthetic drugs are being actively conducted. Silymarin, *Lentinus edodes* mycelia (LEM) extracts, ursodeoxycholic acid (UDCA) and dimethyl diphenyl bicarboxylate (DDB) have been reported as natural products with therapeutic or hepatoprotective effects on ALF [9,10,11,12,13]. Silymarin is an herbal compound derived from *Carduus marianus*, also known as milk thistle. It possesses hepatoprotective effects by inhibiting lipid peroxidation, sustaining membrane permeability from endogenous toxicants and regulating nuclear transcription [9]. *Lentinus edodes* is a medicinal macrofungus showing potential for therapeutic applications in infectious disorders, including hepatitis [10], that exhibits medicinal effects such as anticancer, immunoregulatory and hepatoprotective activities [11]. UDCA is a main compound of dried bile from Chinese black bears and has long been used for therapeutic purposes, especially in liver diseases. It is capable of rendering toxic bile acids nontoxic, which results in cytoprotective, membrane stabilizing, antioxidative and immunomodulatory effects in the liver [12]. DDB is a synthetic analog of Schisandrin C that is isolated from *Schisandrae fructus*. DDB has hepatoprotective effect based on its strong antioxidative ability and has been clinically used to treat viral hepatitis B [13]. Likewise, natural hepatoprotective substances are increasingly being discovered.

*Centella asiatica*, belonging to the family Apiaceae (Umbelliferae), is an herb that is found in most swamps around the world [14]. It is a crucial plant in the international medicinal plant trade for its various ethnomedicinal uses in gastrointestinal disorders, cutaneous troubles and revitalization of brain cells. Several studies have suggested that *C. asiatica* possesses antiulcer [15], antidiabetic [16], cardioprotective [17], radioprotective [18], antimutagenic [19], skin protective [20], immune modulating [21] and memory enhancing [22] effects due to pharmacological properties such as antioxidant [23], antibacterial [24], antifungal [25], antiviral [26], antifilarial [27] and antitumor [28] activities. According to a recent study, *C. asiatica* 50% ethanol extract (CA-HE50) exhibited hepatoprotective effects against acetaminophen (APAP)-induced liver injury [29]. In this study, the hepatoprotective effect of CA-HE50 on ALF is investigated using an LPS/D-Gal-induced ALF animal model. Additionally, the effects of silymarin, LEM, UDCA and DDB, which have already been reported to have a hepatoprotective effect, are investigated together and compared with the effects of CA-HE50.

## 2. Materials and Methods

### 2.1. Animals

Six-week-old C57BL/6 strain mice (*n* = 48) were purchased from OrientBio Co., Ltd. (Seongnam, Gyeonggi, Korea) and were allowed to acclimatize for one week before experiments. Animals determined to be healthy during the acclimatization period were randomly divided into eight groups of six animals each using a Z-shape arrangement method based on body weight. Differences in body weights among groups were confirmed by ANOVA.

Animals were maintained under standard laboratory conditions: temperature of 21.1 ± 1 °C, relative humidity of 49.8 ± 3%, regular photoperiod and illumination (12 h dark/light) and 10–15 times/h ventilation. Animals had free access to standard chow and purified water. The procedures for the experiments and animal care protocol (Approval #: CE20009) were approved by the Internal Animal Care and Use Committee of Chaon Co., Ltd. The experimental procedures followed the National Institutes of Health guidelines for the care and use of laboratory animals.

### 2.2. Treatment

CA-HE50 was supplied by GENENCELL Co., Ltd. The positive control substances, silymarin, LEM, UDCA and DDB, were supplied by Daonpharm Co. Ltd. The eight experimental groups (6 mice per group) were defined as normal control, vehicle control, CA-HE50 (100 and 200 mg/kg) [29], silymarin (100 mg/kg) [30], LEM (200 mg/kg) [31], UDCA (25 mg/kg) [32] and DDB (200 mg/kg) [33]. CA-HE50, silymarin and LEM were dissolved in saline; sonication was performed when they were not well soluble. DDB was dissolved in 0.5% tween 80 in saline and UDCA was dissolved in 0.5% CMC in saline. CA-HE50, silymarin, LEM, UDCA and DDB were administered orally (gavage) every 24 h for 2 weeks. A mixture of LPS (10 µg/kg) and D-gal (200 mg/kg) was injected intraperitoneally 5–6 h before the mice were sacrificed. To maximize the induction of liver toxicity, mice were fasted for 24 h before injection. After respiratory anesthesia with isoflurane, blood was collected and major organs (liver, heart, lung and spleen) were carefully removed. Serum was separated from the collected blood and the separated serum was stored in a deep freezer until used for serum biochemical analyses. The organs were weighed and the liver was cut in half; one half was frozen in liquid nitrogen for the ELISA analysis, stored in a deep freezer, while the other half was fixed in 10% neutral buffered formalin for histopathological examination.

### 2.3. Serum Biochemical Analyses

Serum biochemical analyses were conducted using the collected serum. The activities of biomarkers of liver function (AST, ALT, GGT and LDH) and lipid metabolism biomarkers (TG, HDL, LDL and TC) were measured using an automated serum biochemistry analyzer (Accute, Toshiba, Tokyo, Japan).

### 2.4. Measurement of Oxidative Stress Biomarkers

Half of the collected liver was stored in liquid nitrogen for oxidative stress analysis. The liver samples were washed, minced and homogenized in an assay buffer. The homogenates were centrifuged at 2000× *g* at 4 °C for 10 min and the supernatant was collected. The supernatants were used to measure oxidative stress biomarkers including CAT (Abcam, Cambridge, UK) and SOD (Abcam) with commercial ELISA kits, according to the manufacturers’ instructions.

### 2.5. Histopathological Examination

Liver tissues were fixed in a neutral formalin solution and embedded into paraffin. Paraffin-embedded tissues were sectioned to a thickness of 5 μm using a Reichert 200 microtome (Leica, Nussloch, Eisfeld, Germany). After staining the paraffin sections with hematoxylin and eosin (H&E; Sigma-Aldrich, St. Louis, MO, USA) to evaluate lesions, histopathological changes were examined under a light microscope (Nikon, Tokyo, Japan) and photographs were taken at a magnification of × 200. More precisely, two slides were randomly selected from the liver tissue sections of each mouse; then, three places were randomly selected among each slide and the lesion severity presented per unit area was evaluated. The histopathological analysis scores in the liver parenchyma were calculated under the following standards: 0, as in ‘no lesion’; 1, as in ‘mild injury including focal nuclear pyknosis and cytoplasmic vacuolation’; 2, as in ‘moderate to severe injury showing extensive nuclear pyknosis, cytoplasmic hypereosinophilia and ambiguous boundary between cells’; 3, as in ‘severe with hepatocyte disintegration, congestion and neutrophil infiltration’.

Also, in situ direct DNA fragmentation confirmation was performed using the TUNEL assay commercial kit (Abcam, Cambridge, UK). Fluorescence microscopy was used to visualize TUNEL-positive cells. Two slides were randomly selected from liver tissue sections of each mouse. Then, three places were randomly selected among each slide and the number of apoptotic cells present per unit area was counted.

### 2.6. Statistical Analysis

All experimental results are expressed as mean ± standard deviation (SD) of several independent experiments. Statistical differences were determined by a one-way analysis of variance (ANOVA), followed by Tukey’s multiple comparison test. The GraphPad Prism 5.01 software was used for the statistical analyses.

## 3. Results

### 3.1. Effects of CA-HE50 on Body Weight, Food Consumption and Organ Weight in LPS/D-Gal-induced ALF Animal Model

To confirm the effect of CA-HE50 oral administration for 2 weeks, the changes in body weight, food consumption and weights of major organs were investigated. The body weights of the eight experimental groups were measured on days 0, 8, 14 and 15. There were no significant differences in body weight among all experimental groups throughout the entire process of the experiment (Figure 1A). No clinical symptoms related to body weight were observed regardless of CA-HE50 administration.

Food consumption of each experimental group was measured once a week during the entire experiment. There were no significant differences in food consumption among all experimental groups compared to the vehicle control group (Figure 1B).

The weights of major organs (including heart, liver, lung, spleen and kidney) collected from mice in each group were measured. As shown in Figure 2, compared to the vehicle control group, the heart weight was significantly recovered when treated with 100 and 200 mg/kg CA-HE50 (*p* < 0.05), 100 mg/kg silymarin, 200 mg/kg LEM, 25 mg/kg UDCA and 200 mg/kg DDB (*p* < 0.01). Moreover, the group treated with 100 mg/kg silymarin showed statistically significant increases in kidney weight compared with the vehicle control (*p* < 0.01). Meanwhile, the weights of the liver, lung and spleen were not significantly different among groups.

### 3.2. Effects of CA-HE50 on Liver Damage and Blood Lipids in LPS/D-Gal-Induced ALF Animal Model

We examined the serum levels of AST, ALT, LDH and GGT, which are representative indicators of liver function, to confirm the effect of CA-HE50 on liver function and damage in the LPS/D-Gal-induced ALF animal model. As a result of analyzing AST and ALT in serum, both AST and ALT levels were increased in the vehicle control group compared to the normal group. The AST levels decreased significantly in the CA-HE50, silymarin, LEM, UDCA and DDB administration groups compared to the vehicle control group (*p* < 0.01; Figure 3A). The AST level of the 200 mg/kg CA-HE50 group was not significantly different from that of the positive control (silymarin, LEM, UDCA and DDB) groups.

The ALT levels of the CA-HE50, silymarin, LEM, UDCA and DDB administration groups were also decreased compared to the vehicle control group. In particular, ALT levels significantly decreased in the group administered 200 mg/kg of CA-HE50 (*p* < 0.01) and 200 mg/kg of DDB (*p* < 0.01; Figure 3B). The ALT level of the 200 mg/kg CA-HE50 group was not significantly different from that of the positive control (silymarin, LEM, UDCA and DDB) groups. The GGT level in serum was decreased in the vehicle control group compared to the normal group. The GGT levels in the CA-HE50, silymarin and LEM administered groups were similar to the normal control group and no significant changes were observed in any group (Figure 3C). The LDH level in serum was significantly increased in the vehicle control group compared to the normal group. The LDH level was dramatically decreased in the CA-HE50, silymarin, LEM, UDCA and DDB administration groups compared to the vehicle control group (*p* < 0.01; Figure 3D). In the 200 mg/kg CA-HE50 administration group, the LDH level was reduced to a level similar to that of the LEM and DDB administration groups and was less decreased than that in the silymarin and UDCA administration group.

Additionally, the serum levels of TG, TC, HDL and LDL, which are representative indicators of lipid metabolism, were investigated to confirm the effect of CA-HE50 on lipid metabolism in the LPS/D-Gal-induced ALF animal model. There were no significant differences in the levels of CHOL, TG, HDL and LDL among groups (Figure 3E–H).

### 3.3. Effect of CA-HE50 on Oxidative Stress in LPS/D-Gal-Induced ALF Animal Model

To confirm the effect of CA-HE50 on oxidative stress in the LPS/D-Gal-induced ALF animal model, we investigated the changes in the activities of CAT and SOD, representative antioxidant enzymes in liver tissue. As shown in Figure 4, the vehicle control group, which received only LPS/D-Gal, demonstrated significant decreases in both CAT and SOD levels compared to the normal group (*p* < 0.01). Compared to the vehicle control group, CAT activity was significantly increased only in the UDCA administration group (*p* < 0.05) and SOD activity was significantly increased in the 100 mg/kg CA-HE50, LEM and UDCA administration groups (*p* < 0.05). In the case of CAT activity, there was no significant difference from vehicle control in the CA-HE50, silymarin, LEM and DDB administration groups and it was only significantly increased in the UDCA administration group. In the case of SOD activity, the 100 mg/kg CA-HE50 administration group increased more significantly than the silymarin, LEM and DDB administration groups and it increased less than the UDCA administration group.

### 3.4. Effect of CA-HE50 on Histopathological Changes in LPS/D-Gal-Induced ALF Animal Model

We confirmed the histopathological changes in liver tissue through H&E staining and TUNEL analysis of extracted liver tissue. Histopathological changes in liver tissue were quantified as lesion scores according to the degree of the lesion. In addition, dead cells were counted. The histopathological lesion scores of the vehicle control group were increased significantly compared to the normal control group (*p* < 0.01). We confirmed that the histopathological lesion scores tended to decrease in the CA-HE50 and positive control groups compared to the vehicle control group. In the CA-HE50 administration group, the lesion score decreased more in the 100 mg/kg administration group than in the 200 mg/kg administration group. Among the CA-HE50 and positive control groups, the lesion score in the UDCA administered group (*p* < 0.05) was the most significantly reduced and the CA-HE50 administered group decreased to a level similar to that of the silymarin, LEM and DDB administered groups (Figure 5A). On the other hand, as a result of confirming dead cells through the TUNEL analysis, the numbers of dead cells in the CA-HE50 and positive control-provided groups were significantly smaller in number than those in the vehicle control. (*p* < 0.05). In particular, significance was confirmed at the *p* < 0.001 level in the 200 mg/kg CA-HE50 group (Figure 5B). By checking the level of dead cells, which was not properly confirmed in H&E staining, through the TUNEL analysis, it was confirmed that CA-HE50 is helpful in liver protection.

## 4. Discussion

ALF refers to a rapid loss of hepatocytes. To investigate the hepatoprotective properties of CA-HE50 on ALF, we used an LPS/D-gal-induced ALF animal model. The most common method to induce ALF in animal models is by intraperitoneal injection of a mixture of LPS and D-gal [34,35]. LPS is an endotoxin component of Gram-negative bacteria that binds to a receptor that stimulates inflammatory responses. D-gal is an amino derivative of the sugar galactose, which is frequently used with LPS to induce oxidative stress in hepatocytes and liver inflammation, leading to ALF occurrence [36]. Since the LPS/D-gal-induced ALF model has a different mechanism compared to other drug-induced ALF models, it is frequently used to study the beneficial effects of natural products [37]. Once D-gal is injected, it is metabolized by the Leloir pathway during galactose metabolism to produce reactive oxygen species (ROS) [38]. As ROS deform intracellular proteins, the endoplasmic reticulum (ER) increases autophagy to remove inactive proteins; therefore, it suffers ER stress [39]. This leads to the overexpression of nuclear enzyme poly (ADP-ribose) polymerase (PARP)-1, which depletes uridine triphosphate (UTP) and inactivates intercellular electron transport, thereby reducing mitochondrial protein synthesis [40]. Metabolically active organs such as heart and liver become extremely vulnerable to LPS exposure under such conditions. Meanwhile, as LPS enters the body, cytokine tumor necrosis factor (TNF)-α is induced to combine with TNF receptor type 1 located on the hepatocyte membrane. This up-regulates nuclear factor-kappa B (NF-κB) and intracellular adhesion molecule (ICAM)-1 to induce bursts of inflammation that lead to acute injuries of the liver [41]. In this study, we compare the hepatoprotective properties of CA-HE50 in LPS/D-gal-induced ALF animal models with those of previously known prophylactic drugs, namely, silymarin, LEM, UDCA and DDB.

For comprehensive assessments of LPS/D-gal-induced ALF models, we divide the assessment into four categories: liver function, lipid metabolism, oxidative stress and histopathological analysis. To determine the effects of CA-HE50 and positive controls on liver function in the LPS/D-gal-induced ALF animal model, we measured serum levels of AST, ALT and GGT, which are indicators of liver function. AST is found in organs such as heart, brain, muscle, liver and kidney and is used as a biomarker for damage in these tissues. ALT is an enzyme found in liver muscle cells that is used as a biomarker for liver damage. LDH mediates energy production and is found in liver, kidney and blood cells and may be used as a biomarker for damage in these tissues. GGT is found in bile ducts and is used as a biomarker in hepatocytes for cholestatic liver diseases. The liver plays a central role in lipid metabolism and lipoprotein absorption, formation and export into the circulation. Changes in liver lipid metabolism can contribute to the development of liver diseases such as fatty liver [42]. To confirm the effect of CA-HE50 on lipid metabolism in the ALF animal model, we measured the levels of blood lipids such as TG, TC, LDL and HDL. To confirm histopathological changes in the ALF animal model, we performed histopathological analyses of liver tissue using H&E staining and TUNEL assay. The histopathological lesion scores and count of dead cells were determined based on the damage observed in liver parenchyma cells. Through histopathological analyses, the liver protection efficacy of CA-HE50 and other positive controls (including UDCA and DDB) could be confirmed.

The vehicle control group demonstrated no significant differences in body weight or food consumption, while only heart weight showed significant decreases among organs. According to Wanjun et al., after D-gal treatment, the SOD level in the heart decreased and oxidative stress was enhanced to accelerate the aging of the heart [43]. Thus, a decrease in heart weight indicates weakened cardiomyocytes due to oxidative stress and inflammation. Among the liver function biomarkers, AST, ALT and LDH levels increased compared to the normal control group. Heart and liver tissue damage led to such increases. Both CAT and SOD levels decreased compared to the normal control group. Such CAT and SOD exhaustion indicates that LPS/D-gal induced sudden increase of ROS and inhibited additional CAT and SOD synthesis [36].

Compared to the vehicle control group, the silymarin administration group demonstrated significant differences in heart and kidney weight, AST and LDH levels and histopathological analysis score. Silymarin is known to inhibit membrane permeability and acts as a steroid to regulate the expression of PARP-1, thus reducing the mitochondrial damage caused by ROS [9]. Significant increases in heart weight and decreases in AST levels indicate protection of cardiomyocytes and endothelial cells through such processes. Previous studies revealed the protective effects of silymarin against cardiotoxicity [44]. As PARP-1 is also known to mediate the inflammation pathway [40], we suggest that significant decreases in LDH level were caused by accompanying anti-inflammation effects. No significant differences were found in CAT and SOD levels, which, along with our other results, implies that the hepatoprotective effect of silymarin focuses on building resistance against mitochondrial ROS damage rather than directly removing ROS. In addition, a significant improvement of the histopathological analysis score in the silymarin administration group was observed.

The LEM administration group showed significant differences compared to the vehicle group in heart weight, AST, LDH and SOD levels, as well as in the histopathological analysis scores. The main components of LEM are sugars, proteins, polysaccharides and polyphenolic compounds [45]. LEM is rich in the polyphenol compounds syric acid and vanyl acid and these components have been reported to have hepatoprotective effects by suppressing the expression of α-smooth muscle actin (SMA), a gene related to hepatic fibrosis [11,46,47]. LEM is known to protect liver cells from oxidative stress caused by the CCl_4_-induced production of free radicals by activating the antioxidant enzymes SOD and glutathione peroxidase (GPx) [46]. Our results are consistent with previously reported hepatoprotective effects of LEM and show that LEM has a protective effect against oxidative stress in an LPS/D-gal-induced ALF animal model. 

Compared to the vehicle control group, the UDCA administration group demonstrated significant differences in heart weight, AST, LDH, CAT and SOD levels, while showing significant improvement in histopathological analysis scores. ER stress caused by mitochondrial dysfunction is known to produce toxic bile salt, which possesses detergent-like properties [48]. This toxic bile salt modifies the cellular membrane structure and composition to increase membrane fluidity, leading to apoptosis. UDCA prevents such actions by inhibiting the transition of the apoptosis induction protein Bax from cytosol to mitochondria [49]. Previous studies revealed that the modification of bile acid metabolism caused by liver dysfunction can cause heart defects and that UDCA has protective effects against such actions [50]. UDCA is also known to directly control extracellular signal-regulated kinase (Erk) 1/2 and p38 to restore protein synthesis by removing intracellular ROS while inhibiting inflammation pathways [51]. Anti-oxidative and anti-inflammatory properties of UDCA might result in significant differences in AST, LDH, CAT and SOD levels. However, the lack of difference in ALT levels suggests that such actions are not focused solely on the liver. Among all CA-HE50 administration and positive control groups, only the UDCA group demonstrated significant differences in both CAT and SOD levels. Therefore, the significant improvement of histopathological analysis scores in the UDCA group is due to the anti-inflammatory and antioxidant actions of UDCA.

There were statistically significant differences in heart weight, AST, ALT and LDH levels and histopathological analysis scores in the DDB administration group compared to the vehicle control group. DDB has been reported to have hepatoprotective effects in CCl_4_-, D-gal-, thioacetamide-, prednisolone- and concanavalin A-induced liver injury animal models [52]. DDB activates glutathione-related enzymes in hepatocellular mitochondria, resulting in the enhancement of reduced-glutathione levels [53]. This enhancement leads to effective ROS detoxification and relieves oxidative stress in hepatocytes [54,55]. DDB is capable of down-regulating the transcription of TNF-α, which induces inflammatory responses and leads to apoptosis [56]. Heart diseases are closely related to cardiac and systemic glutathione levels [57]. Our results demonstrate that DDB effectively inhibited heart-weight decrease. It can be deduced that the anti-oxidative effect of DDB influenced not only the liver but also other organs. Furthermore, liver function markers, including AST, ALT and LDH levels, as well as histopathological analysis scores, exhibited significant differences. Our results accord with the previously reported hepatoprotective and anti-oxidative effects of DDB [53,54,55].

Both 100 mg/kg and 200 mg/kg CA-HE50 administration groups demonstrated significant differences in heart weights, AST and LDH levels and histopathological analysis scores compared to the vehicle control group. The AST level of the 200 mg/kg CA-HE50 administration group decreased the most compared to the silymarin, LEM, UDCA and DDB administration groups. In the case of ALT levels, the 200 mg/kg CA-HE50 administration group decreased more than the silymarin, LEM and UDCA administration group, but it decreased less than the 200 mg/kg DDB treatment group. In addition, in the 200 mg/kg CA-HE50 administration group, the LDH level was reduced to a level similar to that of the LEM and DDB administration groups and was less decreased compared to the silymarin and UDCA administration groups. In the case of CAT activity, there was no significant difference from vehicle control in the CA-HE50, silymarin, LEM and DDB administration groups and it was only significantly increased in the UDCA administration group. In the case of SOD activity, the 100 mg/kg CA-HE50 administration group increased more significantly than the silymarin, LEM and DDB administration groups and it increased less than the UDCA administration group. In the case of histopathological lesion score, the lesion score of the UDCA-administered group decreased the most and the CA-HE50-administered group decreased to a level similar to that of the silymarin-, LEM- and DDB-administered groups. The 100 mg/kg CA-HE50 administration group showed significant differences in SOD levels, while the 200 mg/kg CA-HE50 administration group exhibited significant differences in ALT levels. In previous studies, *C. asiatica* was shown to have high antioxidant effects through three pathways: inhibition of linoleic acid peroxidation, superoxide free radical scavenging activity and radical scavenging activity [58,59]. In addition, it has been reported that administration of *C. asiatica* extract has a protective effect due to increased CAT and SOD activity in a dimethylnitrosamine (DMN)-induced liver injury model [60]. CA-HE50 showed a protective effect by inhibiting the expression of inflammatory genes in an APAP-induced liver injury model [29]. Moreover, in a study on the prevention of retinal degeneration of CA-HE50, it was confirmed that CA-HE50 and its active compound, asiaticoside, promote the Nrf2/HO-1 signaling pathway [61]. Judging from the results of these studies, the hepatoprotective efficacy of CA-HE50 shown in the LPS/D-gal-induced ALF animal model is expected to occur through the Nrf2/HO-1 signaling pathway, a well-known antioxidant mechanism.

To summarize the mechanism of action for liver protection of each substance, Silymarin has antioxidant and anti-inflammatory mechanisms [62] and LEM has anti-inflammatory and anti-fibrotic mechanisms [10]. In addition, UDCA has a hepatoprotective effect through an antioxidant mechanism [63] and DDB is known to have a hepatoprotective effect by inhibiting lipid peroxidation [64]. Each mechanism of action is different; therefore, it is not possible to accurately compare them. However, results show that CA-HE50 has a slightly greater or similar level of hepatoprotective effect on ALF compared to the positive controls (silymarin, LEM, UDCA and DDB), which are known to have hepatoprotective effects.

Therefore, this study shows the possibility that CA-HE50 can be used as a raw material for hepatoprotection.

## 5. Conclusions

In this study, we compare the hepatoprotective effects of CA-HE50 in LPS/D-Gal-induced ALF animal models with those of previously known prophylactic drugs, nameley, silymarin, LEM, UDCA and DDB. We verified that silymarin, LEM, UDCA and DDB have hepatoprotective and antioxidative effects against LPS/D-Gal-induced ALF. CA-HE50, with its antioxidative and anti-inflammatory constituents, showed satisfactory results in terms of changes in liver function indicators, oxidative stress indicators and histopathological lesion scores, compared to the positive control groups. These results show that CA-HE50 has a protective effect against ALF induced by LPS/D-Gal and, in particular, helps improve liver function and prevent liver tissue damage caused by oxidative stress. These results suggest that CA-HE50 could be a useful therapeutic drug for the prevention and treatment of ALF.

## Figures and Tables

**Figure 1 nutrients-13-04090-f001:**
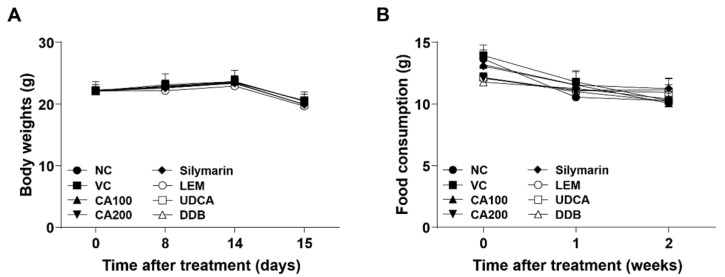
Effects of CA-HE50 on body weight and food consumption changes in LPS/D-Gal-induced ALF animal model. (**A**) The body weights of mice in eight different experimental groups were measured on days 0, 8, 14 and 15. (**B**) Food consumption of each experimental group was measured once per week during the entire experiment. NC, normal control; VC, vehicle control; CA100, 100 mg/kg CA-HE50; CA200, 200 mg/kg CA-HE50; Silymarin, 100 mg/kg silymarin; LEM, 200 mg/kg LEM; UDCA, 25 mg/kg UDCA; DDB, 200 mg/kg DDB. Data are represented as mean ± SD.

**Figure 2 nutrients-13-04090-f002:**
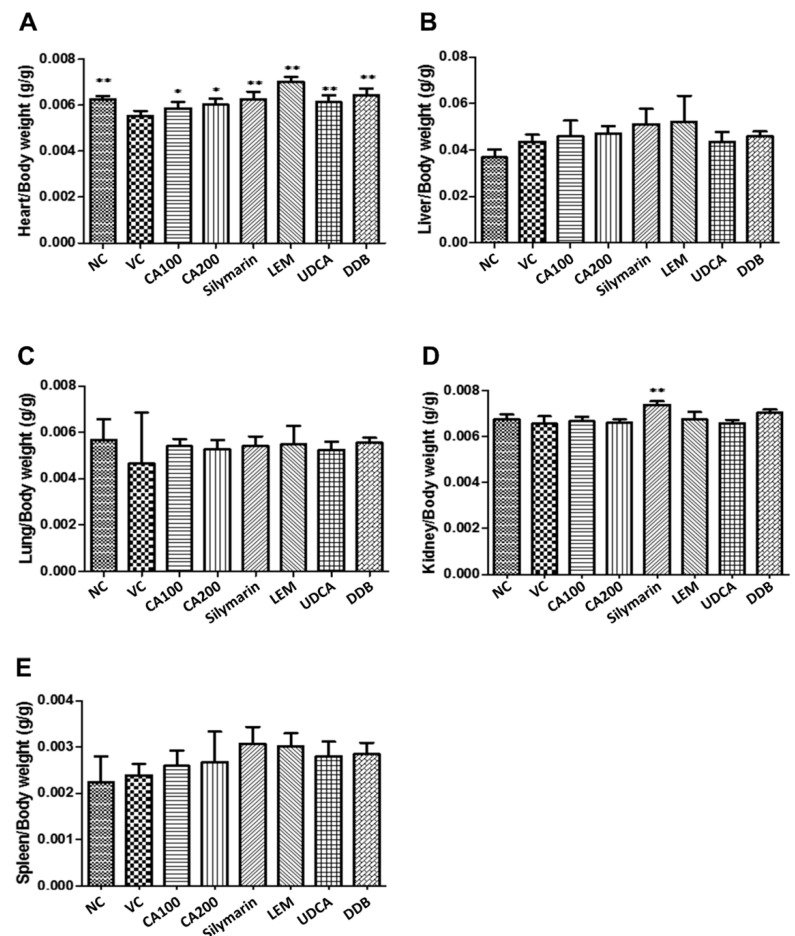
Effects of CA-HE50 on relative organ weight changes in LPS/D-Gal-induced ALF animal model. The weights of major organs collected from mice in each group were measured. (**A**) Heart; (**B**) liver; (**C**) lung; (**D**) spleen; (**E**) kidney. NC, normal control; VC, vehicle control; CA100, 100 mg/kg CA-HE50; CA200, 200 mg/kg CA-HE50; Silymarin, 100 mg/kg silymarin; LEM, 200 mg/kg LEM; UDCA, 25 mg/kg UDCA; DDB, 200 mg/kg DDB. Data are represented as mean ± SD (* *p* < 0.05 or ** *p* < 0.01 vs. vehicle control).

**Figure 3 nutrients-13-04090-f003:**
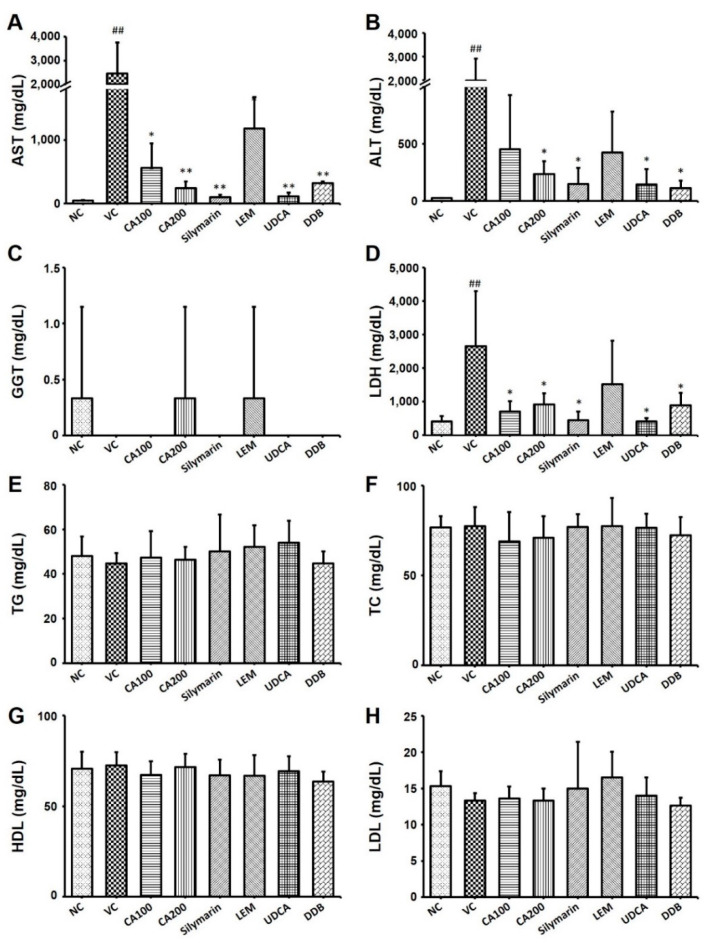
Effects of CA-HE50 on liver damage and blood lipids in LPS/D-Gal-induced ALF animal model. The serum levels of (**A**) AST, (**B**) ALT, (**C**) GGT, (**D**) LDH, (**E**) TG, (**F**) TC, (**G**) HDL and (**H**) LDL were investigated. NC, normal control; VC, vehicle control; CA100, 100 mg/kg CA-HE50; CA200, 200 mg/kg CA-HE50; Silymarin, 100 mg/kg silymarin; LEM, 200 mg/kg LEM; UDCA, 25 mg/kg UDCA; DDB, 200 mg/kg DDB. Data are represented as mean ± SD (^##^
*p* < 0.01 vs. normal control; * *p* < 0.05 or ** *p* < 0.01 vs. vehicle control).

**Figure 4 nutrients-13-04090-f004:**
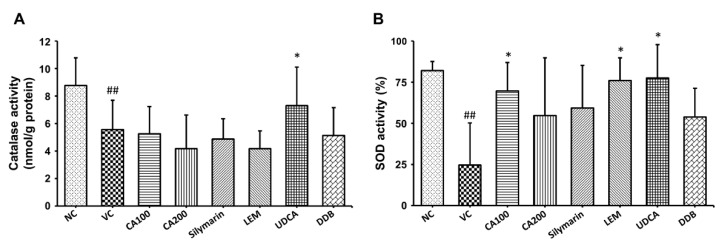
Effects of CA-HE50 on CAT and SOD activities in LPS/D-Gal-induced ALF animal model. The activity changes of representative antioxidant enzymes (**A**) CAT and (**B**) SOD in liver tissue were investigated. NC, normal control; VC, vehicle control; CA100, 100 mg/kg CA-HE50; CA200, 200 mg/kg CA-HE50; Silymarin, 100 mg/kg silymarin; LEM, 200 mg/kg LEM; UDCA, 25 mg/kg UDCA; DDB, 200 mg/kg DDB. Data are represented as mean ± SD (^##^
*p* < 0.01 vs. normal control; * *p* < 0.05 vs. vehicle control).

**Figure 5 nutrients-13-04090-f005:**
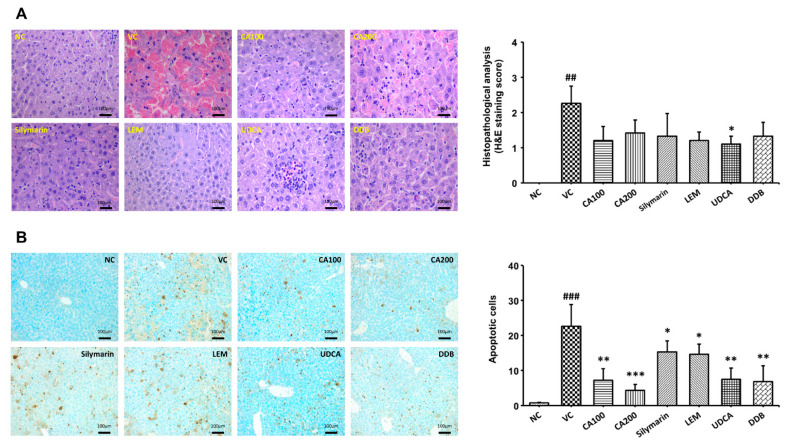
Effect of CA-HE50 on histopathological changes in LPS/D-Gal-induced ALF animal model. Histopathological changes in the extracted liver tissue were confirmed by H&E and TUNEL analysis. Histopathological changes in liver tissue were quantified by lesion scores according to the severity of the lesion. (**A**) Representative images of each group in the histopathological lesion. Histopathological lesion scores are presented as a bar diagram. (**B**) Representative images of each group in apoptotic dead cells by TUNEL analysis. The numbers of apoptotic dead cells are presented as a bar diagram. NC, normal control; VC, vehicle control; CA100, 100 mg/kg CA-HE50; CA200, 200 mg/kg CA-HE50; Silymarin, 100 mg/kg silymarin; LEM, 200 mg/kg LEM; UDCA, 25 mg/kg UDCA; DDB, 200 mg/kg DDB. Data are represented as mean ± SD (^##^
*p* < 0.01 or ^###^
*p* < 0.001 vs. normal control; * *p* < 0.05, ** *p* < 0.01 or *** *p* < 0.001 vs. vehicle control).

## Data Availability

The data presented in this study are available on request from the corresponding author. All data generated or analyzed during this study are included in the manuscript.

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
