# Peer review of "A Comparative Study of the Hepatoprotective Effect of Centella asiatica Extract (CA-HE50) on Lipopolysaccharide/d-galactosamine-Induced Acute Liver Injury in C57BL/6 Mice"

_nutrients, 2021, doi:10.3390/nu13114090_

Round 1
Reviewer 1 Report
In the present study, Hong et al. investigated the plausible hepatoprotective effect of C. asiatica 50% ethanol extract (CA-HE50) on acute liver injury in mouse mediated by Lipopolysaccharide / galactosamine. Authors also compare the observed effect with those of the Lentinus edodes micelia (LEM) extracts, ursodeoxycholic acid (UDCA), and dimethyl diphenyl bicarboxylate (DDB), already proven to be beneficial for the liver.
Author reported that CA-HE50, similarly to silymarin, LEM, UDCA, and DDB has hepatoprotective and antioxidative effects against Lipopolysaccharide / galactosamine, evaluated by AST, ALT, lactate dehydrogenase levels, superoxide dismutase activity and histopathological liver lesions.
I have only minor issues:
Methods: How dosage and time of administration where chosen? How was food consumption measured?
Results: mistake in the figure 3 and 4; please add figure 4
Discussion: line 286-306 should be shortened and reported in the Introduction. Line 428: please remove as you can see.
Reviewer 2 Report
This manuscript tried to address the hepatoprotection effect of CA-HE50 using LPS/D-Gal-induced Acute liver failure mouse model by only accessed ALT/AST level and LDH level and other antioxidant measurement, histopathological scores, and apoptotic cell measurement of TUNEL. However, authors should have address what is underlying mechanism for hepatoprotective effect CA-HE50 in this working model. Moreover, the current version of manuscript is not ready to being reviewed as Figure 3 presented in legend figure 4. Figure 4 (for CAT and SOD activities) completely omitted.
- What is the meaning of heart/body weight ratio changes in ALF model?
- Authors addressed that Centerlla asiatica has a potent effects in antidiabetic, cardioprotective, radioprotective etc.. however did not address what kinds of compounds in CA-HE50 will have a potent effect on ALF model? Or can authors assume what kinds of compounds will be in CA-HE50 to have the hepatoprotective effect?
- In Figure 5, how measure to do histopathological analysis using H&E staining? please address more precisely. CA200 treatment groups looks more liver failure compared to CA100 treatment group in H&E staining however CD200 treatment group showed less apoptosis cells compared to CA100 treatment. Please explain about why CA200 treatment group has less apoptotic cells appeared rather than CA100 treatment group.
- Please address the possible mechanism of CA-HE50 at least in discussion section.
Round 2
Reviewer 2 Report
Happy to accept the current version of manuscript.
This manuscript is a resubmission of an earlier submission. The following is a list of the peer review reports and author responses from that submission.
Round 1
Reviewer 1 Report
In the presented study the Authors evaluated the effect of Centella asiatica ethanolic extract administration on the LPS+D-gal induced acute liver failure in mice model. The study is interesting and well designed, but there are some flaws in results description and interpretation which need to be addressed. These remarks, and some other minor comments are listed below:
Centella asiatica and Lentinus edodes are Latin names of a plant and mushroom, thus they should be written in italics throughout the manuscript.
The abstract should be rewritten, there should not be statistical significances written in the abstract (the p value).
Page 2, line 54 there should probably be “the” before “liver cells”.
Please rewrite the ending of the introduction section. The introduction should not be finished with a conclusion, there should be some kind of hypothesis or the aim of a study. Current version ends with the main results/findings. Please form a hypothesis instead.
In the methods section the Authors wrote that they separated serum from blood and the serum was then used for “blood biochemical analysis”. While they separated the serum from the blood, these analyses were “serum biochemical” not “blood biochemical” analyses. Moreover, there should be “analyses” not “analysis”. In the whole manuscript blood needs to be changed into serum, when results from the serum are described.
How many mice were per one group?
Please explain in more detail the fragment about freezing half of the liver and fixing in formalin: was each liver cut in half and one half was frozen and the other half was fixed? Or maybe whole livers from half of the animals from each group was destined for freezing and from remaining animals for fixing?
As far as results description is concerned: The Author used many positive controls with substances which have proven hepatoprotective activity. But in the analyses they did not describe the differences between analyzed Centella extract and these positive controls. The utilized statistical test allows to compare all groups. E.g. Centella extract at dose of 100 mg/kg with silymarin or DDB etc. Such a description would be informative in order to evaluate whether the studied extract is more or less potent than these substances used as positive controls. It is possible that there will be no differences between groups treated with examined extracts and positive controls, but it needs to be described.
How is it possible that in the graph depicting GGT there are no values for groups G2, G5, G7 and G8?
The Authors wrote in the lines 235-237: “Among the CA-HE50 and positive control groups, the lesion score in the UDCA administered group was the most significantly reduced” – there should be added that in comparison to the vehicle treated group. But once again – the groups should be compared between themselves as it provides a more detailed information on Centella extracts when compared to the positive control-substances. Also the two doses of Centella extract should be compared - one to another.
In the discussion section:
Line 249 there is “gram-negative bacteria”. “Gram-negative” or “Gram-positive” bacteria are named after the last name of a scientist, H.C.J. Gram, who developed the staining, thus “Gram” should be written with a capital letter.
In lines 266-268 it is stated: “In this study, we compared the hepatoprotective properties of CA-HE50 in LPS/D-gal-induced ALF animal models with those of previously known prophylactic drugs: silymarin, LEM, UDCA, and DDB.” – this is not entirely true. The Authors compared these substances only to the non-treated, vehicle group, but not between themselves.
Lines 288-293: “This indicates that cardiomyocytes are weakened due to oxidative stress and inflammation. Among the liver function biomarkers, AST, ALT, and LDH levels increased compared to the normal control group. Heart and liver tissue damage led to such increases. Both CAT and SOD levels decreased compared to the normal control group. Such CAT and SOD exhaustion indicates that LPS/D-gal induced sudden increase of ROS and inhibited additional CAT and SOD synthesis.” – is there any proof in the scientific literature to support these hypotheses?
Regarding the conclusions: “In this study, we compared the hepatoprotective properties of CA-HE50 in LPS/D-gal-induced ALF animal models with those of previously known prophylactic drugs: silymarin, LEM, UDCA, and DDB.” – it must be highlighted once again – there was no statistical comparison provided among these groups. The Authors did not compare results obtained for Centella extract-treated groups with the results obtained for the positive control groups. After reevaluation of the results with regard to statistical comparison between all the groups, this sentence will be valid.
Author Response
Dear, Editor in Chief:
Thank you for reviewing our manuscript and offering your suggestions, which have helped us improve our manuscript greatly. The manuscript has been revised per your comments. Therefore, we are submitting the revised manuscript and a version indicating all the revisions undertaken.
Below, we have included our point-by-point responses to the reviewers’ comments.
Reviewer’s Comments
Reviewer 1: Comments to the authors
In the presented study the Authors evaluated the effect of Centella asiatica ethanolic extract administration on the LPS+D-gal induced acute liver failure in mice model. The study is interesting and well designed, but there are some flaws in results description and interpretation which need to be addressed. These remarks, and some other minor comments are listed below:
- Centella asiatica and Lentinus edodes are Latin names of a plant and mushroom, thus they should be written in italics throughout the manuscript.
Response: We checked whether Centella asiatica and Lentinus edodes were written in italics throughout the manuscript, and parts not written in italics were corrected (lines 19, 22, 530, 531, and 534).
- The abstract should be rewritten, there should not be statistical significances written in the abstract (the p value).
Response: The abstract modified as follows:
Abstract section (lines 18‒38):
Acute liver failure (ALF) refers to the sudden loss of liver function and is accompanied by several complications. In a previous study, we revealed the protective effect of Centella asiatica 50% ethanol extract (CA-HE50) on acetaminophen-induced liver injury. In the present study, we investigated the hepatoprotective effect of CA-HE50 in a lipopolysaccharide/galactosamine (LPS-D-Gal)-induced ALF animal model and compared it to existing therapeutics silymarin, Lentinus edodes mycelia (LEM) extracts, ursodeoxycholic acid (UDCA), and dimethyl diphenyl bicarboxylate (DDB). Serum aspartate aminotransferase (AST) and alanine aminotransferase (ALT) levels were decreased in the CA-HE50, silymarin, LEM, UDCA, and DDB groups compared to the vehicle control group. In particular, AST and ALT levels of the 200 mg/kg CA-HE50 group were significantly decreased compared to positive control groups. Lactate dehydrogenase (LDH) levels were significantly decreased in the CA-HE50, silymarin, LEM, UDCA, and DDB groups compared to the vehicle control group, and LDH levels of the 200 mg/kg CA-HE50 group were similar compared to positive control groups. Superoxide dismutase (SOD) activity was significantly increased in the 100 mg/kg CA-HE50, LEM, and UDCA groups compared to the vehicle control group, and in particular, the 100 mg/kg CA-HE50 group increased significantly compared to positive control groups. In addition, the histopathological lesion score was significantly decreased in CA-HE50 and positive control groups compared with the vehicle control group, and the histopathological lesion score of the 200 mg/kg CA-HE50 group was similar to positive control groups. These results show that CA-HE50 has antioxidant and hepatoprotective effects at a level similar to that of silymarin, LEM, UDCA, and DDB, which are known to have hepatoprotective effects, and CA-HE50 has potential as a prophylactic and therapeutic agent in ALF.
- Page 2, line 54 there should probably be “the” before “liver cells”.
Response: In the revised manuscript, “liver cells” was corrected to “the liver cells” (line 60).
- Please rewrite the ending of the introduction section. The introduction should not be finished with a conclusion, there should be some kind of hypothesis or the aim of a study. Current version ends with the main results/findings. Please form a hypothesis instead.
Response: Reflecting the opinions of reviewers, the ending of the introduction section in the revised manuscript has been rewritten as follows:
Introduction section (lines 92‒96)
In this study, the hepatoprotective effect of CA-HE50 on ALF was investigated using an LPS/D-Gal-induced ALF animal model. Additionally, the effects of silymarin, LEM, UDCA and DDB, which have already been reported to have a hepatoprotective effect were investigated together and compared with the effects of CA-HE50.
- In the methods section the Authors wrote that they separated serum from blood and the serum was then used for “blood biochemical analysis”. While they separated the serum from the blood, these analyses were “serum biochemical” not “blood biochemical” analyses. Moreover, there should be “analyses” not “analysis”. In the whole manuscript blood needs to be changed into serum, when results from the serum are described.
Response: “Blood biochemical analysis” was modified to “Serum biochemical analyses” and “Serum biochemical analyses were” throughout the revised manuscript (lines 126, 130‒131, and 197).
- How many mice were per one group?
Response: Six mice per group were used. In the revised manuscript, the materials and method section (p. 3, lines 114 ~ 117) was revised as follows:
Materials and Methods section (lines 114‒117)
The eight experimental groups (6 mice per group) were defined as normal control, vehicle control, CA-HE50 (100 and 200 mg/kg), silymarin (100 mg/kg), LEM (200 mg/kg), UDCA (25 mg/kg), and DDB (200 mg/kg).
- Please explain in more detail the fragment about freezing half of the liver and fixing in formalin: was each liver cut in half and one half was frozen and the other half was fixed? Or maybe whole livers from half of the animals from each group was destined for freezing and from remaining animals for fixing?
Response: Each liver cut in half and one half was frozen and the other half was fixed. In the revised manuscript, the materials and method section (lines 126‒129) was revised as follows.
Materials and Methods section (lines 126‒129)
The organs were weighed, and each liver was cut in half, and half was frozen in liquid nitrogen for ELISA analysis, stored in a deep freezer, and the other half was fixed in 10% neutral buffered formalin for histopathological examination.
- As far as results description is concerned: The Author used many positive controls with substances which have proven hepatoprotective activity. But in the analyses they did not describe the differences between analyzed Centella extract and these positive controls. The utilized statistical test allows to compare all groups. E.g. Centella extract at dose of 100 mg/kg with silymarin or DDB etc. Such a description would be informative in order to evaluate whether the studied extract is more or less potent than these substances used as positive controls. It is possible that there will be no differences between groups treated with examined extracts and positive controls, but it needs to be described.
Response: We added a comparative explanation between the CA-HE50 and positive control treatment groups for each analysis in the abstract, results and discussion sections of the revised manuscript.
Lines 26‒27: In particular, AST and ALT levels of the 200 mg/kg CA-HE50 group were significantly decreased compared to positive control groups.
Lines 29‒30: and LDH levels of the 200 mg/kg CA-HE50 group were similar compared to positive control groups.
Lines 32‒38: In addition, the histopathological lesion score was significantly decreased in CA-HE50 and positive control groups compared with the vehicle control group, and the histopathological lesion score of the 200 mg/kg CA-HE50 group was similar to positive control groups. These results show that CA-HE50 has antioxidant and hepatoprotective effects at a level similar to that of silymarin, LEM, UDCA, and DDB, which are known to have hepatoprotective effects, and CA-HE50 has potential as a prophylactic and therapeutic agent in ALF.
Lines 203‒205: In particular, the AST level of the 200 mg/kg CA-HE50 administration group decreased the most compared to the silymarin, LEM, UDCA, and DDB administration group.
Lines 216‒218: The ALT level of the 200 mg/kg CA-HE50 administration group decreased more than the silymarin, LEM, and UDCA administration groups, but it decreased less than the 200 mg/kg DDB treatment group.
Lines 225‒228: In the 200 mg/kg CA-HE50 administration group, the LDH level was reduced to a level similar to that of the LEM and DDB administration groups, and was less decreased com-pared to the silymarin and UDCA administration group.
Lines 241‒246: In the case of CAT activity, there was no significant difference from vehicle control in the CA-HE50, silymarin, LEM, and DDB administration groups, and only significantly in-creased in the UDCA administration group. In the case of SOD activity, the 100 mg/kg CA-HE50 administration group increased more significantly than the silymarin, LEM, and DDB administration groups, and it increased less than the UDCA administration group.
Lines 264‒265: and the CA-HE50 administered group decreased to a level similar to that of the silymarin, LEM, and DDB administered group (Fig. 5).
Lines 396‒414: The AST level of the 200 mg/kg CA-HE50 administration group decreased the most compared to the silymarin, LEM, UDCA, and DDB administration group. In the case of ALT level, the 200 mg/kg CA-HE50 administration group decreased more than the silymarin, LEM, and UDCA administration group, but it decreased less than the 200 mg/kg DDB treatment group. In addition, in the 200 mg/kg CA-HE50 administration group, the LDH level was reduced to a level similar to that of the LEM and DDB administration groups, and was less decreased compared to the silymarin and UDCA administration group. In the case of CAT activity, there was no significant difference from vehicle control in the CA-HE50, silymarin, LEM and DDB administration groups, and only significantly increased in the UDCA administration group. In the case of SOD activity, the 100 mg/kg CA-HE50 administration group increased more significantly than the silymarin, LEM, and DDB administration groups, and it increased less than the UDCA administration group. In the case of histopathological lesion score, the lesion score of the UD-CA-administered group decreased the most, and the CA-HE50-administered group de-creased to a level similar to that of the silymarin, LEM, and DDB-administered group. Although it cannot be accurately compared because each mechanism of action is different, these results show that CA-HE50 has a slightly greater or similar level of hepatoprotective effect on ALF compared to silymarin, LEM, UDCA, and DDB, which are known to have hepatoprotective effects.
- How is it possible that in the graph depicting GGT there are no values for groups G2, G5, G7 and G8?
Response: A GGT level of mouse is considered normal when it is 5 or less, and when the value is 0 or not completely zero, but very small, the graph appears as if there is no value. In that case, it appears as if there is no value on the graph. Therefore, it is possible that there are no values for the G2, G5, G7 and G8 groups in the graph representing GGT.
- The Authors wrote in the lines 235-237: “Among the CA-HE50 and positive control groups, the lesion score in the UDCA administered group was the most significantly reduced” – there should be added that in comparison to the vehicle treated group. But once again – the groups should be compared between themselves as it provides a more detailed information on Centella extracts when compared to the positive control-substances. Also, the two doses of Centella extract should be compared - one to another.
Response: We added a comparative explanation between the CA-HE50 and positive control groups for histopathological lesion scores in the results (p. 8, lines 264‒265) and discussion (p. 10, line 408‒410) sections of the revised manuscript.
Line 264‒265: and the CA-HE50 administered group decreased to a level similar to that of the silymarin, LEM, and DDB administered group (Fig. 5).
Line 408‒409: In the case of histopathological lesion score, the lesion score of the UDCA-administered group decreased the most, and the CA-HE50-administered group decreased to a level similar to that of the silymarin, LEM, and DDB-administered group.
Also, in the revised manuscript (p. 8, lines 264‒265), comparisons with vehicle control and between the two doses of CA-HE50 treatment groups have already been mentioned.
Line 264‒265: and the CA-HE50 administered group decreased to a level similar to that of the silymarin, LEM, and DDB administered group (Fig. 5).
- In the discussion section: Line 249 there is “gram-negative bacteria”. “Gram-negative” or “Gram-positive” bacteria are named after the last name of a scientist, H.C.J. Gram, who developed the staining, thus “Gram” should be written with a capital letter.
Response: “gram-negative” was corrected to “Gram-negative” throughout the revised manuscript (line 279 and line 507).
- In lines 266-268 it is stated: “In this study, we compared the hepatoprotective properties of CA-HE50 in LPS/D-gal-induced ALF animal models with those of previously known prophylactic drugs: silymarin, LEM, UDCA, and DDB.” – this is not entirely true. The Authors compared these substances only to the non-treated, vehicle group, but not between themselves.
Response: We added a comparative explanation between the CA-HE50 and positive control groups for each analysis in the abstract, results and discussion sections of the revised manuscript.
Lines 26‒27: In particular, AST and ALT levels of the 200 mg/kg CA-HE50 group were significantly decreased compared to positive control groups.
Lines 29‒30: and LDH levels of the 200 mg/kg CA-HE50 group were similar compared to positive control groups.
Lines 32‒38: In addition, the histopathological lesion score was significantly decreased in CA-HE50 and positive control groups compared with the vehicle control group, and the histopathological lesion score of the 200 mg/kg CA-HE50 group was similar to positive control groups. These results show that CA-HE50 has antioxidant and hepatoprotective effects at a level similar to that of silymarin, LEM, UDCA, and DDB, which are known to have hepatoprotective effects, and CA-HE50 has potential as a prophylactic and therapeutic agent in ALF.
Lines 203‒205: In particular, the AST level of the 200 mg/kg CA-HE50 administration group decreased the most compared to the silymarin, LEM, UDCA, and DDB administration group.
Lines 216‒218: The ALT level of the 200 mg/kg CA-HE50 administration group decreased more than the silymarin, LEM, and UDCA administration groups, but it decreased less than the 200 mg/kg DDB treatment group.
Lines 225‒228: In the 200 mg/kg CA-HE50 administration group, the LDH level was reduced to a level similar to that of the LEM and DDB administration groups, and was less decreased com-pared to the silymarin and UDCA administration group.
Lines 241‒246: In the case of CAT activity, there was no significant difference from vehicle control in the CA-HE50, silymarin, LEM, and DDB administration groups, and only significantly in-creased in the UDCA administration group. In the case of SOD activity, the 100 mg/kg CA-HE50 administration group increased more significantly than the silymarin, LEM, and DDB administration groups, and it increased less than the UDCA administration group.
Lines 264‒265: and the CA-HE50 administered group decreased to a level similar to that of the silymarin, LEM, and DDB administered group (Fig. 5).
Lines 396‒414: The AST level of the 200 mg/kg CA-HE50 administration group decreased the most compared to the silymarin, LEM, UDCA, and DDB administration group. In the case of ALT level, the 200 mg/kg CA-HE50 administration group decreased more than the silymarin, LEM, and UDCA administration group, but it decreased less than the 200 mg/kg DDB treatment group. In addition, in the 200 mg/kg CA-HE50 administration group, the LDH level was reduced to a level similar to that of the LEM and DDB administration groups, and was less decreased compared to the silymarin and UDCA administration group. In the case of CAT activity, there was no significant difference from vehicle control in the CA-HE50, silymarin, LEM and DDB administration groups, and only significantly increased in the UDCA administration group. In the case of SOD activity, the 100 mg/kg CA-HE50 administration group increased more significantly than the silymarin, LEM, and DDB administration groups, and it increased less than the UDCA administration group. In the case of histopathological lesion score, the lesion score of the UD-CA-administered group decreased the most, and the CA-HE50-administered group de-creased to a level similar to that of the silymarin, LEM, and DDB-administered group. Although it cannot be accurately compared because each mechanism of action is different, these results show that CA-HE50 has a slightly greater or similar level of hepatoprotective effect on ALF compared to silymarin, LEM, UDCA, and DDB, which are known to have hepatoprotective effects.
- Lines 288-293: “This indicates that cardiomyocytes are weakened due to oxidative stress and inflammation. Among the liver function biomarkers, AST, ALT, and LDH levels increased compared to the normal control group. Heart and liver tissue damage led to such increases. Both CAT and SOD levels decreased compared to the normal control group. Such CAT and SOD exhaustion indicates that LPS/D-gal induced sudden increase of ROS and inhibited additional CAT and SOD synthesis.” – is there any proof in the scientific literature to support these hypotheses?
Response: In the revised manuscript, we have added to the discussion section (lines 318‒321 and 325) the following texts and references to support these hypotheses. The effects of LPS and D-Gal are mentioned in lines 282 to 294 of the discussion section.
Discussion section (lines 318‒321)
According to Wanjun et al., after D-gal treatment, the SOD level in the heart decreased and oxidative stress was enhanced to accelerate the aging of the heart. [39]. Thus, a decrease in heart weight indicates weakened cardiomyocytes due to oxidative stress and inflammation.
Additional scientific literatures (line 319 and 325)
- Wanjun, M.; Shanshan, W.; Weijun, P.; Taoli, S.; Jianhua, H.; Rong Y.; Bikui, Z.; Wenqun, L. Antioxidant effect of Polygonatum sibiricumpolysaccharides in D-Galactose-Induced heart aging mice. Biomed Res Int 2021, 2021, 6688855.
- Seyed-Mahdi, M.; Tourandokht, B.; Davood, N.; Ali, M.K.; Mehrdad, R. Protective effect of diosgenin on LPS/D-Gal-induced acute liver failure in C57BL/6 mice. Microb Pathog 2020, 146, 104243.
- Regarding the conclusions: “In this study, we compared the hepatoprotective properties of CA-HE50 in LPS/D-gal-induced ALF animal models with those of previously known prophylactic drugs: silymarin, LEM, UDCA, and DDB.” – it must be highlighted once again – there was no statistical comparison provided among these groups. The Authors did not compare results obtained for Centella extract-treated groups with the results obtained for the positive control groups. After reevaluation of the results with regard to statistical comparison between all the groups, this sentence will be valid.
Response: We performed between all the groups statistical analyzes again. In the revised manuscript, the descriptions in the statistical analysis (see p. 4, lines 157‒159) and figure legend sections (see p. 5, lines 192‒194; p. 6, lines 211‒212; p. 7, line 252; p. 8, line 273‒274) have been revised as follows:
Lines 157‒159: (#p<0.05, ##p<0.01 vs. normal control; *p<0.05, **p<0.01, ***p<0.001 vs. vehicle control; &p<0.05, &&p<0.01, &&&p<0.001 vs. 200 mg/kg CA-HE50).
Lines 192‒194: Data are represented as mean ± SD (##p<0.01 vs. normal control; *p<0.05 or **p<0.01 vs. vehicle control; &p<0.05, &&p<0.01 or &&&p<0.001 vs. 200 mg/kg CA-HE50).
Lines 211‒212: Data are represented as mean ± SD (#p<0.05 vs. normal control; *p<0.05 or **p<0.01 vs. vehicle control; &p<0.05 vs. 200 mg/kg CA-HE50).
Line 252: #p<0.05 vs. normal control
Line 273‒274: ##p<0.01 vs. normal control
Also, we added a comparative explanation between the CA-HE50 and positive control groups for each analysis in the abstract, results and discussion sections of the revised manuscript as follows:
Lines 26‒27: AST and ALT levels of the 200 mg/kg CA-HE50 group were significantly decreased compared to positive control groups.
Lines 29‒30: and LDH levels of the 200 mg/kg CA-HE50 group were similar compared to positive control groups.
Lines 32‒38: In addition, the histopathological lesion score was significantly decreased in CA-HE50 and positive control groups compared with the vehicle control group, and the histopathological lesion score of the 200 mg/kg CA-HE50 group was similar to positive control groups. These results show that CA-HE50 has antioxidant and hepatoprotective effects at a level similar to that of silymarin, LEM, UDCA, and DDB, which are known to have hepatoprotective effects, and CA-HE50 has potential as a prophylactic and therapeutic agent in ALF.
Lines 203‒205: In particular, the AST level of the 200 mg/kg CA-HE50 administration group decreased the most compared to the silymarin, LEM, UDCA, and DDB administration group.
Lines 216‒218: The ALT level of the 200 mg/kg CA-HE50 administration group decreased more than the silymarin, LEM, and UDCA administration groups, but it decreased less than the 200 mg/kg DDB treatment group.
Lines 225‒228: In the 200 mg/kg CA-HE50 administration group, the LDH level was reduced to a level similar to that of the LEM and DDB administration groups, and was less decreased compared to the silymarin and UDCA administration group.
Lines 241‒246: In the case of CAT activity, there was no significant difference from vehicle control in the CA-HE50, silymarin, LEM, and DDB administration groups, and only significantly increased in the UDCA administration group. In the case of SOD activity, the 100 mg/kg CA-HE50 administration group increased more significantly than the silymarin, LEM, and DDB administration groups, and it increased less than the UDCA administration group.
Lines 264‒265: and the CA-HE50 administered group decreased to a level similar to that of the silymarin, LEM, and DDB administered group (Fig. 5).
Lines 396‒414: The AST level of the 200 mg/kg CA-HE50 administration group decreased the most compared to the silymarin, LEM, UDCA, and DDB administration group. In the case of ALT level, the 200 mg/kg CA-HE50 administration group decreased more than the si-lymarin, LEM, and UDCA administration group, but it decreased less than the 200 mg/kg DDB treatment group. In addition, in the 200 mg/kg CA-HE50 administration group, the LDH level was reduced to a level similar to that of the LEM and DDB administration groups, and was less decreased compared to the silymarin and UDCA administration group. In the case of CAT activity, there was no significant difference from vehicle control in the CA-HE50, silymarin, LEM and DDB administration groups, and only significantly increased in the UDCA administration group. In the case of SOD activity, the 100 mg/kg CA-HE50 administration group increased more significantly than the silymarin, LEM, and DDB administration groups, and it increased less than the UDCA administration group. In the case of histopathological lesion score, the lesion score of the UD-CA-administered group decreased the most, and the CA-HE50-administered group de-creased to a level similar to that of the silymarin, LEM, and DDB-administered group. Although it cannot be accurately compared because each mechanism of action is different, these results show that CA-HE50 has a slightly greater or similar level of hepatoprotective effect on ALF compared to silymarin, LEM, UDCA, and DDB, which are known to have hepatoprotective effects.

Reviewer 2 Report
The manuscript submitted by Hong et al. describes a study evaluating the potential therapeutic benefit of an herbal extract on acute-on chronic liver failure (ALF), due to the authors' previous study demonstrating CA-HE50 protected form acetaminophen-related liver injury. The study is relevant to public health and thus worthy of investigation. It is well written and easy to understand. Upon review, the manuscript requires significant revision and additional data to maximize the impact of the study's findings and to ensure scientific rigor and reproducibility. Listed below are my recommendation and comments.
1- General: The study needs more data to be presented and better characterization of the models and data shown
2- Figures 1 and 2 are akin to supplemental data and should be combined and included separate from the main text.
3- The coded nature of the data is difficult to interpret and is too much for the reader. The axes should be labeled with shorthand abbreviations for the treatments and possibly split into ALF with CA-HE50 versus the other interventions. A figure legend should also be included for each.
4- There are formatting issues in the figures that might be only due to the proof as presented. Ensure appropriate resolution and that all words/axes are able to be viewed.
5- Describing as ALF-induced is a bit of a stretch. It is better to say that LPS/D-Gal is a model of ALF.
6- The variation in Figure 3, especially with ALT/AST/GGT/LDH are quite high. Despite being SD, even if converted to SEM would still be high. It suggests that there were some non-responders in these treatment groups. Increased experimental replicates should be included in a revision.
7- There are no histopathology slides shown, just scoring. Nor are there relevant endpoints to liver injury such as TUNEL staining.
8- As LPS/D-gal is a model of ALF that involves the immune system, what is the mRNA expression of inflammatory genes and characterization of the immune infiltrate?
9- Some in vitro modeling of the effect of CA-HE50 protection on hepatocytes and mechanistic studies should be included.
Author Response
Please see our 'response to reviewer' file.

Round 2
Reviewer 1 Report
The Authors have improved the results description, still, I have some concerns.
Line 203-205 (which has been added): “In particular, the AST level of the 200 mg/kg CA-HE50 administration group decreased the most compared to the silymarin, LEM, UDCA, and DDB administration group.” – there are no statistical markings (&) indicating that 200 mg/kg CA-HE50 has decreased the most compared to positive controls. The same goes for ALT measurement – there is an asterisk above the bar of CA200 and DDB indicating that there is probably no difference between them, moreover, there are no “&” markings above the other positive control bars – it cannot be concluded that “200 mg/kg CA-HE50 administration group decreased more than the silymarin, LEM, and UDCA administration groups, but it decreased less than the 200 mg/kg DDB treatment group”. This would be true if there were “&” markings above bars for silymarin, LEM and UDCA.
I think there is some kind of misunderstanding here. The Authors compare the statistical markings from the “wrong side”. It seems to me that the Authors read the statistical markings in the figures as follows: The CA-HE50 200 mg/kg possesses asterisk and has relatively low bar in the graph and the positive controls have slightly higher bars with no asterisk, thus CA-HE50 has greater impact than positive controls which have no statistical marking compared to vehicle control. In fact as long as there is no “&” marking (chosen by the Authors to depict differences between CA-HE50 200 mg/kg group and positive controls) it cannot be said that CA-HE50 having an asterisk has a greater effect than positive controls which have no asterisks. It can be considered as greater/stronger/weaker effect ONLY if the p value between CA-HE50 and the silymarin or LEM or UDCA or DDB is lower than 0.05. Otherwise, if p is equal or higher than 0.05 there is no statistically significant difference between CA-HE50 200 mg/kg and positive control substances.
Other thing which has caught my attention – why only the dose 200 mg/kg was compared with positive controls? What about the 100 mg/kg dose? It should also have its own statistical markings when compared to positive control groups.
I suggest changing the markings completely - into letters. Add letters above the bars – the groups which are statistically insignificant should share the same letter. I am not familiar with the GraphPad Prism 5.01 software and thus I do not know if it allows to obtain such differentiation between groups, but such a marking is more readable. But from what I have found in the literature databases, it can be concluded that there are published works, in which the Authors used GraphPad Prism 5.01 software and also used the letters as statistical markings (e.g. “Logarithmic Transformation is Essential for Statistical Analysis of Fungicide EC50 Values” by Liang et al., ” Baseline Sensitivity and Toxic Actions of Prochloraz to Sclerotinia sclerotiorum” by Zhang et al., “Peripheral modulation of olfaction by physiological state in the Egyptian leaf worm Spodoptera littoralis (Lepidoptera: Noctuidae)” by Martel et al. or “Regulatory Roles of Cytokinins and Cytokinin Signaling in Response to Potassium Deficiency in Arabidopsis” by Nam et al.)
I shall provide a fictive example:
Group 1 – letter above a high bar: “a”
Group 2 – letters above a small bar: “cd”
Group 3 – letters above a medium bar: “bc”
Group 4 – letters above a medium bar: “ab”
How it should it be described: The values in the groups 2 and 3 were significantly lower than in the group 1. The value in the group 4 was significantly higher than in the group 2. The values obtained for group 4 were not statistically different from the values obtained in groups 1 and 3. Values recorded in the groups 2 and 3 were statistically insignificant.
Letter markings are easier to follow and provide more information.
Responses to my other remarks are satisfactory.
Author Response
Please confirm the file the 'response to reviewer.'

Reviewer 2 Report
I thank the authors for a quick turnaround to the previous review. Unfortunately, the revised manuscript and responses to the first review are inadequate to improve the impact and novelty of this study as well as its suitability for publication. My issues with the revision and what is lacking are listed below.
1- The variation still presented in this study suggests an effect that might not be reproducible in future studies. More data from existing samples could provide more evidence that the experimental variation is not an issue. As was mentioned in the first review.
2- The lack of additional data to validate the LPS/D-gal model and protective/therapeutic effect of CA-HE50 in the hands of this investigative team is further troubling. Inflammatory gene expression, TUNEL staining, etc.
3- The additional histopathology of scoring does very little to confirm the previous data. It just shows a scale used for scoring, not labeled with treatments or showing representative images with CA-HE50, silymarin, UDCA, etc.
4- No in vitro/mechanistic data, primary cells.
